# Semantic Positive Pairs for Enhancing Visual Representation Learning of Instance Discrimination Methods

**Mohammad Alkhalefi**                                    *m.alkhalefi1.21@abdn.ac.uk*
*Department of Computing Science*
*University of Aberdeen, UK*

**Georgios Leontidis**                                    *georgios.leontidis@abdn.ac.uk*
*Department of Computing Science & Interdisciplinary Centre for Data and AI*
*University of Aberdeen, UK*

**Mingjun Zhong**                                    *mingjun.zhong@abdn.ac.uk*
*Department of Computing Science*
*University of Aberdeen, UK*

**Reviewed on OpenReview:** *https://openreview.net/forum?id=z5AXLMBWdU*

## Abstract

Self-supervised learning algorithms (SSL) based on instance discrimination have shown promising results, performing competitively or even outperforming supervised learning counterparts in some downstream tasks. Such approaches employ data augmentation to create two views of the same instance (i.e., positive pairs) and encourage the model to learn good representations by attracting these views closer in the embedding space without collapsing to the trivial solution. However, data augmentation is limited in representing positive pairs, and the repulsion process between the instances during contrastive learning may discard important features for instances that have similar categories. To address this issue, we propose an approach to identify those images with similar semantic content and treat them as positive instances, thereby reducing the chance of discarding important features during representation learning and increasing the richness of the latent representation. Our approach is generic and could work with any self-supervised instance discrimination frameworks such as MoCo and SimSiam. To evaluate our method, we run experiments on three benchmark datasets: ImageNet, STL-10 and CIFAR-10 with different instance discrimination SSL approaches. The experimental results show that our approach consistently outperforms the baseline methods across all three datasets; for instance, we improve upon the vanilla MoCo-v2 by 4.1% on ImageNet under a linear evaluation protocol over 800 epochs. We also report results on semi-supervised learning, transfer learning on downstream tasks, and object detection.

## 1 Introduction

In supervised learning, models are trained with input data $X$ and their corresponding semantic labels or classes $Y$. It is rather common for each class to have several hundreds of instances available for training, which enables the model to extract the important features and create useful representations for the given samples (Van Gansbeke et al., 2020). This type of machine learning has been proven to perform well in various domains whenever data are available in abundance. In practice, that is often not the case, given that data annotation is laborious and expensive.

Recently, self-supervised learning (SSL) algorithms based on instance discrimination reduced the reliance on large annotated datasets for representation learning (He et al., 2020; Chen et al., 2020a;b; Misra & Maaten,

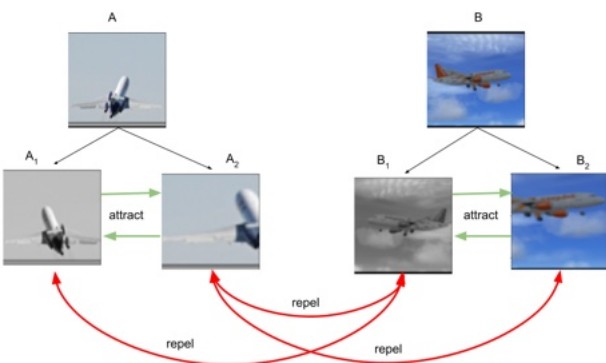

Figure 1: Example of an instance discrimination task where positive pairs are attracted together and negative pairs are pushed apart, even if they have similar semantic content.

2020; Chen & He, 2021; Grill et al., 2020). These approaches treat each instance as a class on its own, so they employ random data augmentation for every single image in the dataset to prevent the model from learning trivial features and become invariant to all augmentations (Tian et al., 2020; Xiao et al., 2020; Chen et al., 2020a; Misra & Maaten, 2020). The model learns the image representation by attracting the positive pairs (i.e., two views of the same instance) closer in the embedding space without representation collapse. Contrastive and non-contrastive instance discrimination methods (Chen et al., 2020a; Caron et al., 2020; He et al., 2020; Grill et al., 2020; Bardes et al., 2021) have shown promising performance, often close to supervised learning or even better in some downstream tasks (Chen et al., 2020a; Chen & He, 2021; Chuang et al., 2020; Misra & Maaten, 2020; Huynh et al., 2022; Dwibedi et al., 2021). However, these approaches have two main limitations: 1) the data augmentation is limited in representing positive pairs (i.e., can not cover all variances in given class); 2) the repulsion process between the images in contrastive learning is implemented regardless of their semantic content, which may cause the loss of important features and slow model convergence (Huynh et al., 2022). For example, Figure 1 shows two images that have similar semantic content (aeroplanes). In contrastive instance discrimination, the images are repelled in the embedding space because the objective is to attract positive pairs and push apart all other images, despite their similar semantic content. This is a major limitation that requires attention, as the performance of the downstream tasks depends on high-quality representations learnt by self-supervised pre-training (Donahue et al., 2014; Manová et al., 2023; Girshick et al., 2014; Zeiler & Fergus, 2014; Durrant & Leontidis, 2022; Kim & Walter, 2017; Durrant & Leontidis, 2023).

Two recent prominent approaches, which are Nearest-Neighbor Contrastive Learning of Visual Representations (NNCLR) (Dwibedi et al., 2021), and False Negative Cancellation (FNC) (Huynh et al., 2022) proposed to find semantic pairs in the dataset, treating them as positive pairs during representation learning, to improve the model performance on downstream tasks. NNCLR uses a support set $Q$ to keep a representation of the dataset during model training. The model learns visual representations by creating a representation for the two views of the input instance $(z_i, z_i^+)$, and then finds the nearest neighbours for the first view (i.e., $z_i$) from the support set $Q$, and so the nearest neighbour is $S = NN(z_i, Q)$. NNCLR then treats $S$ as semantic positive pairs for the second view of the instance (i.e., $z_i^+$) regardless of the semantic content of $S$, which may belong to a category different to $z_i^+$. FNC creates more than two views for each instance which are $[z_i, z_i^+, z_s^1, z_s^2]$, where $(z_i, z_i^+)$ are positive pairs and $(z_s^1, z_s^2)$ are support views for the instance. They find the potential semantic pairs for the support views of each instance from negative examples in the batch by computing their similarity. Finally, they treat the identified semantic pairs as positive pairs during model training.

Finding images that have similar content (i.e., same category) and treating them as positive pairs increases the data diversity which could thus improve the power of representation learning (Tian et al., 2020; Dwibedi et al., 2021; Huynh et al., 2022; Khosla et al., 2020). Contrariwise, mapping wrong semantic pairs and

encouraging the model to treat them as positive pairs causes reduced representation learning and slow model convergence. In this paper, we introduce an approach for enhancing the process of finding semantic pairs to improve visual representation learning. We name our method Semantic Positive Pairs for enhancing Instance Discrimination (SePP- ID) since our experiments indicate that accurate semantic positive pairs could significantly improve the performance of instance discrimination SLL. For identifying semantic positive pairs, we propose to use pre-trained models to firstly map the original images from the dataset (i.e., not augmented images) into latent representations and then the semantic positive pairs are matched by computing the similarity scores based on the latent representation vectors. Any self-supervised learning algorithms could be used as a pre-trained model for our purpose. For example, our experiments show that a model pre-trained by MoCo-v2 (Chen et al., 2020b) approach is good for identifying semantic positive pairs. These semantic positive pairs set (SPPS) along with the positive pairs (i.e., two views for the same instance) are used to train instance discrimination self-supervised model. The key difference to our approach is using pre-trained models and original images from the dataset to match semantic positive pairs, whereas the methods mentioned above, such as NNCLR and FNC are finding semantic positive pairs during the representation learning. Such approaches may have a lower chance of matching the right semantic positive pairs, leading to slow model convergence and degrading representation learning. One possible reason would be that the model requires the necessary number of epochs to converge to learn a similar representation for the images from the same category (i.e., at the beginning of model training the semantic pairs are mapped based on texture and color, later in training the model become better in recognize class); another reason would be that the similarity score was computed using the embedding vectors of the augmented images. For example, with 1000 epochs, NNCLR reached 57% accuracy in terms of true semantic pairs, while FNC achieved 40% accuracy (Dwibedi et al., 2021; Huynh et al., 2022). Also, we show an empirical example in Figure 2 that using augmented images and a non-pre-trained model to find semantic pairs may lead to mapping wrong semantic pairs. In contrast, the accuracy in terms of true semantic positive pairs was achieved over 92% when MoCo-v2 was used as the pre-trained model with the original dataset in our proposed approach. Our contributions are as follows:

- We propose to use the representations of the pre-trained models for the original images from the dataset to match semantic positive pairs, which achieved as high as 92% accuracy in terms of true semantic positive pairs, compared to 57% using NNCLR and 40% using FNC.

- We demonstrate that SePP-ID outperformed other state-of-the-art (SOTA) approaches. For example, SePP-ID achieved 76.3% accuracy on ImageNet, compared to 74.4% using FNC.

- We demonstrate that the SPPS found by using our scheme improve the performance of several instance discrimination approaches across various epoch scenarios and datasets which indicates that the SPPS could be adapted to improve visual representation learning of any other instance discrimination approach.

## 2 Related Work

Several SSL approaches have been proposed, all of which aim to improve representation learning and achieve better performance on a downstream task. In this section, we provide a brief overview of some of these approaches, but we would encourage the readers to read the respective papers for more details.

**Clustering-Based Methods:** The samples that have similar features are assigned to the same cluster. Therefore, discrimination is based on a group of images rather than on instance discrimination (Caron et al., 2020; Van Gansbeke et al., 2020; Caron et al., 2018; Asano et al., 2019). DeepCluster (Caron et al., 2018) obtains the pseudo-label from the previous iteration which makes it computationally expensive and hard to scale. SWAV (Caron et al., 2020) solved this issue by using online clustering, but it needs to determine the correct number of prototypes. Also, there has been a body of research in an area called Multi-view Clustering (MVC) (Yang et al., 2022a; Trosten et al., 2021; Lu et al., 2023; Li et al., 2022a) whereby contrastive learning is used with clustering to find semantic pairs and alleviate the false negative issue. However, our approach does not rely on a clustering method to identify semantic pairs, providing two main advantages over clustering approaches. Firstly, we avoid some drawbacks of clustering methods,

such as predefining the number of prototypes (i.e. clusters), dealing with less separable clusters, and relying on labels or pseudo-labels. Secondly, our approach can be applied to both contractive and non-contrastive learning, as demonstrated in this work.

**Distillation Methods:** BYOL (Grill et al., 2020) and SimSiam (Chen & He, 2021) use techniques inspired by knowledge distillation where a Siamese network has an online encoder and a target encoder. The target network parameters are not updated during backpropagation. Instead, the online network parameters are updated while being encouraged to predict the representation of the target network. Although these methods have produced promising results, it is not fully understood how they avoid collapse. Self-distillation with no labels (DINO) (Caron et al., 2021) was inspired by BYOL but the method uses a different backbone (ViT) and loss function, which enables it to achieve better results than other self-supervised methods while being more computationally efficient. Bag of visual words (Gidaris et al., 2020; 2021) also uses a teacher-student scheme, inspired by natural language processing (NLP) to avoid representation collapse. The student network is encouraged to predict the features' histogram for the augmented images, similar to the teacher network's histogram.

**Information Maximization:** Barlow twins (Zbontar et al., 2021) and VICReg (Bardes et al., 2021) do not require negative examples, stop gradient or clustering. Instead, they use regularisation to avoid representation collapse. The objective function of these methods aims to reduce the redundant information in the embeddings by making the correlation of the embedding vectors closer to the identity matrix. Though these methods provide promising results, they have some limitations, such as the representation learning being sensitive to regularisation. The effectiveness of these methods is also reduced if certain statistical properties are not available in the data.

**Contrastive Learning:** Instance discrimination, such as SimCLR, MoCo, and PIRL (Chen et al., 2020a; He et al., 2020; Chen et al., 2020b; Misra & Maaten, 2020) employ a similar idea. They attract the positive pairs together and push the negative pairs apart in the embedding space albeit through a different mechanism. SimCLR (Chen et al., 2020a) uses an end-to-end approach where a large batch size is used for the negative examples and both encoders' parameters in the Siamese network are updated together. PIRL (Misra & Maaten, 2020) uses a memory bank for negative examples and both encoders' parameters are updated together. MoCo (Chen et al., 2020b; He et al., 2020) uses a moment contrastive approach whereby the query encoder is updated during backpropagation and the query encoder updates the key encoder. The negative examples are located in a dictionary separate from the mini-batch, which enables holding large batch sizes.

**Enhanced Contrastive Learning:** Such mechanisms focus on the importance of the negative examples and find different ways to sample negative examples regardless of their content, which may cause undesired behaviour between images with similar semantic content. Some studies have focused on improving the quality of the negative examples which in turn improves representation learning. (Kalantidis et al., 2020) and (Robinson et al., 2020) focused on the hard negative samples around the positive anchor, whereas (Wu et al., 2020) introduced the percentile range for negative sampling. Another approach introduced by Chuang et al.(Chuang et al., 2020) gives weights for positive and negative terms to reduce the effects of undesirable negatives. Other methods such as (Dwibedi et al., 2021), (Huynh et al., 2022), and (Auh et al., 2023) use similarity metrics to define the images that have similar semantic content and treat them as positive pairs during model training. Although these approaches provide a solution to determine variant instances for the same category and treat them as positive pairs, there are common drawbacks in these approaches that hinder them from mapping highly accurate semantic pairs, such as:

1. They use a non-pre-trained model to represent the images before measuring the similarity between them.

2. They compute the similarity between transformed images, not the original images in the original dataset.

Counting on non-pre-trained models with augmented images to find semantic positive pairs akin to (Dwibedi et al., 2021; Huynh et al., 2022; Auh et al., 2023) may lead to inaccurate results. To demonstrate such issue, Figure 2 shows an empirical example of the inaccurate similarity scores we obtained when we used a non-pre-trained model and random augmented images to determine the instances that belong to the same class

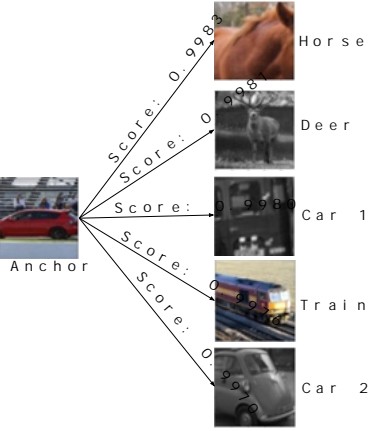

Figure 2: Similarity scores are shown for anchor (car) with instances from other classes using non-pre-trained models and random augmented images.

in the dataset. The similarity scores show that the horse and deer are more similar to the anchor (i.e., car) than car 1 and car 2, and the train is more similar to the anchor than car 2, which is not correct either.

**Semi-Supervised Learning:** The advantage of training a model on instance variants of the same category to improve the representation learning is used in semi-supervised approaches (Bošnjak et al., 2023; Yang et al., 2022b). (Bošnjak et al., 2023) present an approach to train the model on semantic positive pairs by leveraging a few labelled data. In their approach, they use labelled data and k-nearest neighbour to provide pseudo-labels for unlabelled points based on the most frequent class near the unlabelled data. Following that, they treat the data points that have similar pseudo-labels as semantic positive pairs in a contrastive learning setting. Such methods still require labelled data during training to provide semantic positive pairs. Our proposed pre-processing method provides a different way of approaching this, as we will demonstrate below across several datasets and ablation studies. We used a model pre-trained with SSL approaches and worked with the original dataset rather than augmented images to determine semantic positive pairs. In addition, our method does not require labelled data, specialised architecture or support set to hold the semantic positive pairs, which makes it easier to be integrated with any self-supervised learning methods.

## 3 Methodology

This section proposes an approach to enhancing the visual representation learning of instance discrimination SSL methods by using semantic positive pairs (i.e., two different instances belong to the same category). To achieve that, we introduce the Semantic Sampler whose purpose is to find semantic positive pairs from the training data. The main idea of our approach is to find semantic positive pairs set (SPPS) from the original dataset (i.e., not distorted images by augmentation) by using the Semantic Sampler which consists of a pre-trained model and similarity metric. As shown in Figure 3 for finding the SPPS, the pre-trained model in the Semantic Sampler firstly maps $K$ images from the original dataset into latent representations and then the semantic positive pairs are matched by computing the similarity scores based on the latent representation vectors. Note that $K$ is a constant number that defines the number of images involved in the process of finding semantic pairs. These SPPS are used along with positive pairs from the dataset for training instance discrimination SSL models. In the literature, as noted previously, there are methods using semantic pairs to improve the performance of instance discrimination SSL, such as the FNC (Huynh et al., 2022), the NNCLR and (Dwibedi et al., 2021). In this section, we propose a different scheme for searching SPPS, inducing a modified loss function for training contrastive instance discrimination model. Our experiments show that our approach significantly improves the performance of instance discrimination methods and outperforms the SOTA approaches including both FNC and NNCLR. In the following, we will introduce how the SPPS is obtained as well as our SSL algorithm using SPPS. Algorithm 1 shows how the proposed method is implemented.

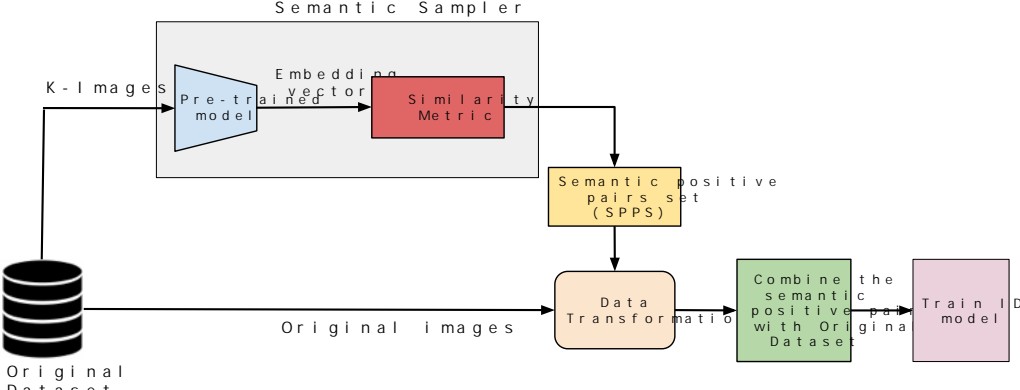

Figure 3: The proposed methodology: Firstly, $k$ images are chosen from the dataset and encoded by the pre-trained model; Secondly, a similarity metric is used to find the semantic positive pairs for each anchor, followed by data transformations applied to both the original dataset and the semantic positive pairs set. Eventually, all the images are combined in one dataset which will be used to train the instance discrimination model.

### 3.1  Semantic Positive Pairs

We use a pre-trained SSL model with similarity metrics in the Semantic Sampler to search for semantic positive pairs in the original dataset. Figure 4 shows the whole process for creating SPPS.

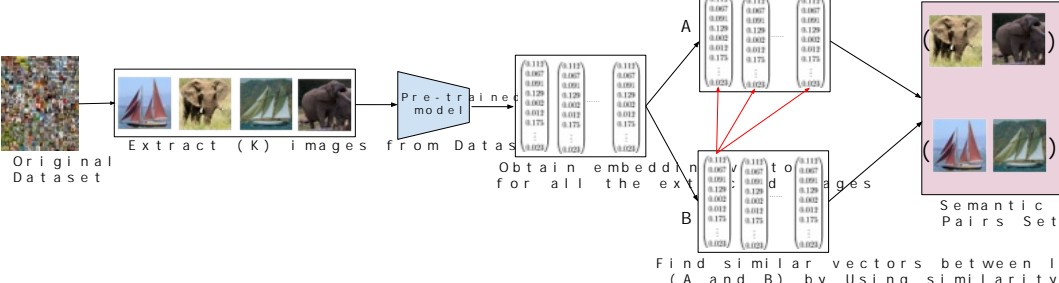

Figure 4: Illustrate the process of Semantic Sampler in identifying the semantic positive pairs.

As shown in Figure 4, $K$ images are randomly chosen from the training data set, and then they are encoded by using the pre-trained SSL model. The embedding vectors generated by the pre-trained model are duplicated in two lists: one list contains anchors and another list contains semantic candidate embedding vectors. In our approach, the cosine similarity function is used to find semantic positive pairs for each anchor (i.e., the list $B$ in Figure 4) from those semantic candidates embedding vectors (i.e., the list $A$ in the figure). The anchors may have more than one positive sample chosen from the candidates. This allows the model to go beyond a single positive instance and capture divergence feature information from different instances belonging to the same category. Eventually, a Semantic positive pair set (SPPS) is created containing pairs of the semantic positive samples. The semantic positive samples were paired when the similarity score lies in the range $[0.97, 0.99]$ and our experiments show that the accuracy of mapping semantic pairs as well as model performance increased on downstream tasks when choosing such threshold values (see Section 4 for computing the accuracy of finding SPPS). Note that the maximum threshold, i.e., 0.99, was chosen to avoid identical image samples. Compared to the previous approaches such as FNC and NNCLR, the semantic positive pairs are found by the Semantic Sampler before training the instance discrimination model, therefore the model is trained on accurately identified semantic pairs from the beginning and this leads to faster convergence and improves the representation learning. On the contrary, FNC and NNCLR search the semantic positive

samples in the batch during the training by using augmented images (i.e., distorted images) and a non-pre-trained model which slows model convergence and degrades representation learning because of inaccurate semantic pairs.

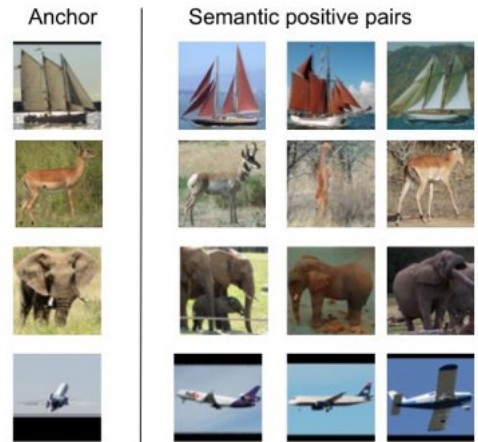

Figure 5: Example of semantic positive pairs found by our approach for different anchors in the STL10-unlabeled dataset.

Figure 5 shows examples of SPPS obtained from the STL10 dataset. It shows that our approach finds the true semantic positive pairs very well for those anchors, despite the anchor image having different properties (e.g. colour, direction, background, and size).

---

**Algorithm 1** Combining SPPS with original data

**Input:** Dataset samples, constant k, and structure *f*.

```
 1: ImageList1=[ ]
 2: ImageList2=[ ]
 3: for k ← 1 to N do
 4:     image ← tensor(x_k)                                    ▷ convert k images from dataset to tensor
 5:     emb_vector ← f(image)                                  ▷ encode images by pre-trained model
 6:     norm_vector ← Normalize(emb_vector)                    ▷ l2 norm
 7:     ImageList1.append(norm_vector)
 8: end for
 9:
10: ImageList2 ← ImageList1                                    ▷ both lists have similar embedding vectors
11: Max= 0.99 and Min= 0.97                                    ▷ define threshold
12: sim= torch.mm(ImageList1, ImageList2.T)                    ▷ compute similarity

13: semantic_pairs_list=[ ]
14: for i ← 1 to sim.size()[0] do
15:     for j ← 1 to sim.size()[1] do
16:         if Sim[i,j] ≥ Min and Sim[i,j] ≤ Max then
17:             Positive_pair ← tuple(Dataset[i], Dataset[j])
18:             semantic_pairs_list.append(Positive_pair)
19:         end if
20:     end for
21: end for
22: applies a random transformation to semantic_pairs_list.
23: combines the semantic_pairs_list with the original dataset.
```

**Output:** combine dataset

---

## 3.2 Learning with Semantic Positive Pairs

This subsection describes how our approach SePP-ID (i.e., Semantic Positive Pairs for enhancing Instance Discrimination) uses SPPS to train instance discrimination SSL models. As demonstrated in Figure 6 a copy for each instance in the original dataset is created and random augmentations are applied for each copy to generate positive pairs. For those semantic positive pairs described in Section 3.1, we do not need to create

copy for the instances because we already have pairs so we only need to apply random transformations to each instance in the pairs. After that, the SPPS is merged with the original training dataset and so the training dataset is slightly larger than the original one.

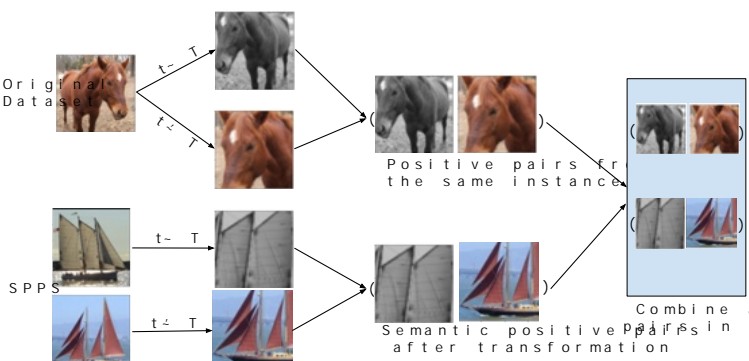

Figure 6: The second step of the methodology: the instances of the original dataset and the SPPS are transformed and combined into one dataset.

After combining the SPPS with the original dataset, we train the contrastive instance discrimination model on the new *combined* dataset with a momentum encoder approach (similar to MoCo-v2 Chen et al. (2020b)). The contrastive loss function attracts the two views of the same instance (i.e., positive pairs) closer in the embedding space while pushing apart all the other instances (i.e., negative samples):

$$\ell(u, v) = -\log \frac{\exp(\text{sim}(u, v)/\tau)}{\sum_{k=1}^{2N} \blacksquare_{[k \neq i]} \exp(\text{sim}(u, w_k)/\tau)} \tag{1}$$

As shown in Equation 1, our approach encourages the model to increase the similarity between the two views $\text{sim}(u, v)$ and reduce the similarity with all other images in the batch $w_k$ where $v$ and $u$ could be either semantic positive pairs (i.e., two images belong to the same category) or positive pair (i.e., two views for the same instance). Note that the number of semantic positive samples of instance $x$ may vary. For example, some instances may not have any semantic positive pair, while others may have more than one semantic positive sample. Thus, the overall loss of our contrastive instance discrimination approach is given by the following equation:

$$loss = \frac{1}{N} \sum_{i=1}^{N} \left[ \ell(u, v) + \sum_{m=1}^{M} \lambda_{im} \ell(u, v_m) \right], \tag{2}$$

where $0 \leq \lambda \leq 1$ represents regularization. In the overall loss function (Equation 2), the contrastive loss for the positive pairs is defined by the first term $\ell(u, v)$ where the second term $\ell(u, v_m)$ defines the semantic positive pairs for the instance that has semantic pairs. In the case of $\lambda_{im} = 0$, there are no semantic positive pairs and the model will be trained only on the positive pairs $\ell(u, v)$, thus the overall loss is equal to the loss of the positive pairs. Under this scenario, the model training is reduced to the original approach such as vanilla MoCo-V2 (Chen et al., 2020b). On the other hand, if $\lambda_{im} = 1$ that means we have semantic pairs so the term for semantic positive pairs $\ell(u, v_m)$ is added to the overall loss and is computed in the same manner that is being used to compute positive pairs. This approach would increase the richness of the latent space during visual representation learning. Our experiments show that this scheme improves model performance on the downstream tasks.

## 4 Experiments and Results

### 4.1 Main Tasks

**Datasets:** We evaluated SePP-ID approach on three datasets, i.e. STL-10 "unlabeled" with 100K training images (Coates & Ng, 2011), CIFAR-10 with 50K training images (Krizhevsky, 2009), and ImageNet-1K with 1.28M training images (Russakovsky et al., 2015).

**Pre-trained model:** Many candidate approaches can be used to train the pre-trained model of the Semantic Sampler for finding SPPS from the original dataset. As a pre-trained model is only used for finding SPPS, we prefer choosing models with relatively low computational overhead. The following are the intuitions on how the approach was chosen to train the Semantic Sampler's pre-trained model: 1) The model backbone was chosen to be ResNet50 because it is used in most instance discrimination approaches so that we can easily compare them; 2) The approach should provide reasonable performance when they are trained on a small batch size (e.g., 256) and a small number of epochs (e.g., 100) to use low computational overhead; 3) The approach uses a small projection head (e.g., 256) while training the model because we want to keep the projection head in the process of creating SPPS. If the projection head has a large output dimension such as VICReg (i.e., 8192 dimensions), it will add a large computational overhead for finding SPPS.

Based on the above intuition, SimSiam (Chen & He, 2021), MoCo-V2 (Chen et al., 2020b) and SimCLR (Chen et al., 2020a) were chosen as the candidate to train the Semantic Sampler's pre-trained model. To choose the best approach for training the pre-trained model across these three approaches, we trained three models on ImageNet-1K for 100 epochs with batch size 256. After model training, we froze the model parameters and kept the projection head (256D). we evaluate the three pre-trained models (i.e., model pre-trained by SimSiam, MoCo-v2, and SimCLR) by creating SPPS and computing the accuracy of selected semantic pairs by leveraging the ImageNet labels. Table 1 demonstrates the performance of the three candidate Semantic Samplers' pre-trained model, showing that MoCo-v2 was the best-performing model for finding SPPS in terms of accuracy. Thus, we utilize the model pre-trained using the MoCo-v2 approach in the Semantic Sampler for all subsequent experiments.

Table 1: The accuracy of finding semantic positive pairs in ImageNet across the three candidate approaches.

| Approach | Accuracy |
|---|---|
| MoCo-v2 | 92.03% |
| SimSiam | 91,44% |
| SimCLR | 90.72% |

**Training Setup:** We use ResNet50 as a backbone and the model is trained with SGD optimizer, weight decay 0.0001, momentum 0.9 and initial learning rate 0.03. The mini-batch size is 256 and the model is trained up to 800 epochs. We set the K-value = 10% of the ImageNet dataset (i.e., 128K random images are picked from the Imagenet dataset for finding semantic pairs). In our ablation study, we compared the performance of our models when various proportions (i.e., various K-value) of data were used for choosing SPPS.

**Evaluation:** We evaluated SePP-ID approach by using linear evaluation, semi-supervised setting, transfer learning and object detection against leading SOTA approaches. In linear evaluation, we followed standard evaluation protocol (Chen et al., 2020a; He et al., 2020; Huynh et al., 2022; Dwibedi et al., 2021). We trained a linear classifier for 100 epochs on top of a frozen backbone pre-trained by SePP-ID. We used an ImageNet training set with random cropping and random left-to-right flipping augmentations to train the linear classifier from scratch. The results are reported on the ImageNet evaluation set with centre crop $(224 \times 224)$. In a semi-supervised setting, we fine-tune the network with 60 epochs using 1% labeled data and 30 epochs using 10% labeled data. Also, we employ a linear evaluation to assess the learned features from the ImageNet dataset on small datasets using transfer learning. Finally, we evaluate the transferability of the learned embeddings by finetuning the model on PASCAL VOC object detection (Everingham et al., 2010).

**Comparing with SOTA Approaches:** We use linear evaluation to compare our approach (i.e., SePP-ID), which is momentum contrastive with semantic positive pairs, against vanilla MoCo-v2 on different epochs on the ImageNet-1k dataset. In addition, we compare our approach after 800 epochs with the performance of other SOTA on the ImageNet-1k.

Table 2: Comparisons between vanilla MoCo-v2 and SePP-ID on the ImageNet dataset with different epochs.

| Approach\Epochs | 100 | 200 | 400 | 800 |
|---|---|---|---|---|
| MoCo-v2 (Chen et al., 2020b) | 67.4% | 69.9% | 71.0% | 72.2% |
| SePP-ID (*proposed*) | **69.2%** | **72.3%** | **75.2%** | **76.3%** |

Table 2 shows that our approach consistently improved the performance of the instance discrimination approach across various numbers of epochs. Our approach SePP-ID significantly outperforms the vanilla MoCo-v2 by 4.1% on 800 epochs. The results substantiate the importance of semantic pairs in instance discrimination representation learning. For example, our approach with 400 epochs surpasses the vanilla MoCo-v2 with 800 epochs by 3%.

Table 3: Comparisons between SePP-ID and SOTA approaches on ImageNet.

| Approach | Epochs | Batch size | Accuracy |
|---|---|---|---|
| MoCo-v2 (Chen et al., 2020b) | 800 | 256 | 72.2% |
| BYOL (Grill et al., 2020) | 1000 | 4096 | 74.4% |
| SimCLR (Chen et al., 2020a) | 1000 | 4096 | 69.3% |
| SimSiam (Chen & He, 2021) | 800 | 512 | 71.3% |
| VICReg (Bardes et al., 2021) | 1000 | 2048 | 73.2% |
| SWAV (Caron et al., 2020) | 800 | 4096 | 75.4% |
| OBoW (Gidaris et al., 2021) | 200 | 256 | 73.8% |
| DINO (Caron et al., 2021) | 800 | 1024 | 75.3% |
| Barlow Twins (Zbontar et al., 2021) | 1000 | 2048 | 73.2% |
| CLSA (Wang & Qi, 2022) | 200 | 256 | 73.3% |
| HCSC (Guo et al., 2022) | 200 | 256 | 73.3% |
| SNCLR (Ge et al., 2023) | 800 | 4096 | 75.3% |
| SCFS (Song et al., 2023) | 800 | 1024 | 75.7% |
| UniVIP (Li et al., 2022b) | 300 | 4096 | 74.2% |
| IFND (Chen et al., 2021) | 200 | 256 | 69.7% |
| CLFN (Auh et al., 2023) | 100 | 512 | 59.6% |
| NNCLR (Dwibedi et al., 2021) | 1000 | 4096 | 75.5% |
| FNC (Huynh et al., 2022) | 1000 | 4096 | 74.4% |
| SePP-ID(*with MoCo-v2, proposed*) | 800 | 256 | **76.3%** |
| SimCLR (Chen et al., 2020a) | 1000 | 256 | 67% |
| NNCLR (Dwibedi et al., 2021) | 1000 | 256 | 68.7% |
| SePP-ID(*with SimCLR*) | 800 | 256 | **69.3%** |

Table 3 highlights the advantage of using our approach to enhance the contrastive instance discrimination SSL approaches, clearly outperforming all baselines. This supports our hypothesis that we can obtain more accurate semantic positive pairs by using Semantic Sampler and the original dataset. Consequently, we can train the instance discrimination models on correct semantic pairs, thereby enhancing representation learning and improving model performance on the downstream task. Also, the result shows that using a non-pre-trained model with augmented images to determine the semantic pairs may slow the model convergence because the model needs several epochs to be able to pick the correct semantic positive pairs. Therefore, our approach achieves 76.3% after 800 epochs, which is better than NNCLR and FNC by 0.8% and 1.9%, respectively (after 1000 epochs).

NNCLR and FNC are attracting the nearest neighbour of the anchor in the embedding space because they assume that they have similar content (same category). However, using a non-pre-trained model and augmented images (distortion images) to find images that have similar content may cause obtaining wrong semantic pairs during model training which leads to slow model convergence and reduced model performance as shown in Table 3. On the contrary, our approach acquires more accurate semantic pairs by using the Semantic Sampler with the original dataset. Therefore, the instance discrimination model is trained on the right semantic pairs from the beginning of the training which leads to improved model performance and fast convergence.

In addition, we do further analysis by using the end-to-end mechanism (i.e., SePP-ID(*with SimCLR*) where both encoders are updated in the backpropagation similar to NNCLR, FNC, and SimCLR. In our experiment, we used batch size 256 because larger batch sizes require memory that exceeds the GPU memory available. The result shows that our approach performs better than both (i.e., NNCLR and SimCLR) when both approaches use 256 as batch size.

**Semi-Supervised Learning on ImageNet:** In this part, we evaluate the performance of SePP-ID under the semi-supervised setting. Specifically, we use 1% and 10% of the labeled training data from ImageNet-1k for fine-tuning, which follows the semi-supervised protocol in SimCLR (Chen et al., 2020a). The top-1 accuracy is reported in Table 4 after fine-tuning using 1% and 10% training data. SePP-ID outperforms all the compared methods. The results demonstrate that SePP-ID achieves the best feature representation quality.

Table 4: Semi-supervised learning results on ImageNet. Top-1 performances are reported on fine-tuning a pre-trained ResNet-50 with ImageNet 1% and 10% datasets.

| Approach\Fraction | top-1 | |
|---|---|---|
| | ImageNet 1% | ImageNet 10% |
| SimCLR (Chen et al., 2020a) | 48.3% | 65.6% |
| BYOL(Grill et al., 2020) | 53.2% | 68.8% |
| SWAV (Caron et al., 2020) | 53.9% | 70.2% |
| DINO (Caron et al., 2021) | 50.2% | 69.3% |
| SCFS (Song et al., 2023) | 54.3% | 70.5% |
| NNCLR (Dwibedi et al., 2021) | 56.4% | 69.8% |
| FNC (Huynh et al., 2022) | 63.7% | 71.1% |
| SePP-ID (*proposed*) | **64.2%** | **71.8%** |

**Transfer Learning on Downstream Tasks:** We follow the linear evaluation setup described in (Grill et al., 2020; Chen et al., 2020a; Dwibedi et al., 2021) to show the effectiveness of transfer representations learned by SePP-ID on multiple downstream classification tasks. The datasets used in this benchmark are as follows: CIFAR Krizhevsky (2009), Stanford Cars Krause et al. (2013), Oxford-IIIT Pets Parkhi et al. (2012), and Birdsnap Berg et al. (2014). We first train a linear classifier using the training set labels while choosing the best regularization hyper-parameter on the respective validation set. Then we combine the training and validation set to create the final training set, which is used to train the linear classifier that is evaluated on the test set.

Table 5: Transfer learning results from ImageNet with the standard ResNet-50 architecture.
*  denotes the results are reproduced in this study.*

| Approach | CIFAR-10 | CIFAR-100 | Car | Birdsnap | Pets |
|---|---|---|---|---|---|
| MoCo-v2 Chen et al. (2020b)* | 91.2% | 74.8% | 51.2% | 43.8% | 84.5% |
| NNCLR (Dwibedi et al., 2021) | 93.7% | 79.0% | 67.1% | 61.4% | 91.8% |
| FNC (Huynh et al., 2022) | 93% | 76.8% | **68.8%** | 54.0% | 89.0% |
| SePP-ID (*proposed*) | **94.5%** | **79.7%** | **68.8%** | **62%** | **92.3%** |

Table 5 presents transfer learning results where our approach SePP-ID improves over all the counterpart approaches on five datasets. This demonstrates that our model learns useful semantic features, enabling it to generalize to unseen data in different downstream tasks.

**Object Detection Task:** To further evaluate the transferability of the learned representation, we fine-tune the model on PASCAL VOC object detection. We use similar settings as in MoCo (Chen et al., 2020b), where we finetune on the VOC trainval07+12 set using Faster R-CNN with a R50-C4 backbone and evaluate on VOC test2007. we fine-tuned for 24k iterations ($\approx 23$ epochs). Table 6 reveals that our approach performs similarly to FNC and better than vanilla MoCo-v2.

Table 6: Transfer learning on Pascal VOC object detection.

| Approach | AP50 |
|---|---|
| MoCo-V2 (Chen et al., 2020b) | 82.5% |
| FNC (Huynh et al., 2022) | 82.8% |
| SePP-ID (*proposed*) | 82.8% |

## 4.2 Ablation Study

In this section, we provide a more in-depth analysis of our approach by conducting four different studies as follows: 1) We apply our method to different datasets to show that our approach performs consistently across different datasets; 2) We experimentally explore the impact of the parameter $k$ on model performance; 3) We also randomly selected images from the original dataset, and then add them into the original data after augmentation to ensure that the improvement of performance was due to the SPPS rather than simply increasing data size; and 4) Conduct experiments on different instance discrimination approaches.

### 4.2.1 Comparisons on Different Datasets

we aim to verify that our method maintains its performance across various datasets when using different backbones (e.g. ResNet18). To achieve this, we compare vanilla MoCo-v2 with our approach, i.e. SePP-ID on two datasets (STL-10 and CIFAR-10) with k-value = 10% of the dataset. Note the K-value with ImageNet dataset is 10%, fix the proportion (10%) with the other datasets (STL-10 and CIFAR-10) to see if we can obtain the same improvement in representation learning with the same proportion on smaller datasets. Table 7 illustrates that our method outperforms the vanilla approach by 4.55% after 200 epochs on STL-10 and achieved 80.36% after 400 epochs which is better than Vanilla MoCo-v2 at 800 epochs. On CIFAR-10, SePP-ID outperforms MoCo-v2 across all the different epochs and achieved 77.46% after 800 epochs which is higher than the vanilla MoCo-v2 by 3.58%. These results show that our approach is working properly even with smaller datasets and different backbones.

Table 7: MoCo-v2 versus SePP-ID on CIFAR-10 and STL-10 with ResNet18.

| Approach\Epoch | STL-10 | | | CIFAR-10 | | |
|---|---|---|---|---|---|---|
| | 200 | 400 | 800 | 200 | 400 | 800 |
| MoCo-v2 | 69.46% | 77.37% | 80.08% | 65.27% | 69.50% | 73.88% |
| SePP-ID (*proposed*) | **74.01%** | **80.36%** | **82.29%** | **72.05%** | **73.87%** | **77.46%** |

### 4.2.2 Impact of the $k$ Value

In this subsection, we experimentally analysed the impact of the $k$ parameter on our model's performance. To obtain the effect of the ($k$-value), we trained models for 100 epochs on the ImageNet dataset and fixed all the parameters except the $k$ parameter. Then, we linearly evaluated the models to see the effect of $k$ on downstream tasks.

Table 8: SePP-ID performance with varying $k$-value.

| $k\%$ | 0% | 5% | 10% | 25% | 50% | 80% | 100% |
|---|---|---|---|---|---|---|---|
| Number of SPPS | 0 | 12,752 | 18,109 | 23,968 | 48,308 | 54,710 | 61,200 |
| Hours for training | $\approx 30$ | $\approx 37$ | $\approx 45$ | $\approx 66$ | $\approx 103$ | $\approx 148$ | $\approx 172$ |
| Accuracy | 67.40% | 68.63% | 69.20% | 69.56% | 69.70% | 69.82% | 69.85% |

It is obvious in Table 8 that when increasing $k$, the number of semantic positive pairs is increased (i.e., Number of SPPS), and so is the model performance. This suggests that semantic positive pairs affect positively the performance of instance discrimination models. On the other hand, a larger $k$ needs more training hours as expected. Therefore, we chose $k$ equal to (10%) for all of our experiments. Note that we used two A100 80GB to train the models in our experiments.

To ensure that the enhancement in the performance was attributed to the semantic positive pairs rather than increasing the size of the dataset, we randomly picked 18109 images from ImageNet (which is similar to the number of semantic positive samples found with 10% of the ImageNet dataset) and created two copies of them ($x_i$ and $x_i'$) each of which is randomly augmented. We added these samples to the original ImageNet dataset. This allows us to test whether the improvement came from the semantic positive pairs or from increasing the size of the dataset. The results shown in Table 9 indicate that simply adding random augmented images to the original dataset has negligible improvement (i.e., 0.2%) to MoCo-v2's performance, while the performance improved by 4.1% when adding semantic positive samples.

Table 9: Comparing the performance of MoCo-V2 with random add augmented images against SePP-ID (k=10%) after 800 epochs on ImageNet.

| Dataset pre-process | Number of added pairs | Accuracy |
|---|---|---|
| MoCo-V2 | 0 | 72.2% |
| MoCo-V2 (added random augmented images) | 18,109 | 72.4% |
| SePP-ID($k = 10\%$) | 18,109 | 76.3% |

### 4.2.3 Non-Contrastive Learning with our Approach

In these experiments, we want to study the effect of our approach on non-contrastive learning. To do so, we used the framework of three SOTA approaches, SimSiam(Chen & He, 2021), DINO (Caron et al., 2021), and VICReg(Bardes et al., 2021).

Table 10: Comparing our approach (SePP-ID) with SimSiam framework versus vanilla Simsiam .

| Approach\Epochs | 100 | 200 | 400 | 800 |
|---|---|---|---|---|
| SimSiam (Chen & He, 2021) | 68.1% | 70.0% | 70.8% | 71.3% |
| SePP-ID (*proposed*) | 68.9% | 71.1% | 71.9 % | 72.5 % |

Table 10 shows that semantic positive pairs found by our approach consistently improve the representation learning of the non-contrastive instance discrimination method (knowledge distillation). our approach surpasses vanilla SimSiam by 1.2% on 800 epochs. Also, our approach with 400 epochs achieved better than the vanilla approach with 800 epochs.

Furthermore, our approach also significantly improves the performance of an information maximisation instance discrimination method, i.e. VICReg. Table 11 indicates that using the VICReg framework with semantic positive pairs found by our approach increases the performance by 2.35% on 100 epochs and 2.9% on 1000 epochs.

Table 11: Comparisons between VICReg and SePP-ID on ImageNet.

| Approach\Epochs | 100 | 1000 |
|---|---|---|
| VICReg (Bardes et al., 2021) | 68.6 % | 73.2% |
| SePP-ID (*proposed*) | 70.95% | 76.1% |

Table 12: Comparing the performance accuracy between vanilla DINO and SePP-ID on ImageNet.

| Approach\Epochs | 200 |
|---|---|
| DINO (Caron et al., 2021) | 66.8% |
| SePP-ID (*proposed*) | 68.6% |

Finally, we used DINO (Caron et al., 2021) frameworks, which use centring and sharping to avoid representation collapse. We used the publicised implementation of DINO to train two models: Vanilla DINO and DINO with semantic positive pairs found by our approach for 200 epochs on the ImageNet dataset. We employed the same hyperparameters used in the original paper. The reported results in Table 12 are for training without multi-crop. It is apparent that semantic positive pairs found by our approach play an important role in improving the representation learning for the DINO framework. Our approach with the DINO framework outperforms the Vanilla DINO by 1.8% in linear evaluation.

## 5 Conclusion

In this paper, we have proposed to use a Semantic Sampler (pre-trained model and similarity metric) with the original dataset to find semantic positive pairs. We demonstrated that these semantic positive samples can significantly improve the visual representation learning of the SSL instance discrimination methods on three datasets which are ImageNet, STL-10, and CIFAR-10. Our experiments also indicate that our approach outperformed other methods, including NNCLR, FNC and CLFN, which are using semantic positive samples. This suggests that the true semantic positive samples are important for learning visual representations with instance discrimination approaches.

## Acknowledgments

We would like to thank the University of Aberdeen's HPC facility for enabling this work.

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
