# OpenReview forum: "Semantic Positive Pairs for Enhancing Visual Representation Learning of Instance Discrimination Methods"
_TMLR — Accepted by TMLR_

### Review · Reviewer_Tg1e · 2023-12-26

**Summary Of Contributions:**

This paper designs an approach named SePP-ID for achieving false-negative robust contrastive learning. Concretely, SePP-ID randomly select K% samples from the dataset as positive candidates and use MoCo-v2 to infer the semantically-relevant pairs in the set. After that, SePP-ID treat these pairs as positives and perform contrastive learning. To verify the effectiveness of the proposed approach, the authors conduct extensive experiments on three widely-used benchmarks.

**Audience:**

No

**Claims And Evidence:**

Yes

**Requested Changes:**

Please see the weaknesses

**Strengths And Weaknesses:**

Strengths:
This paper devises a new false-negative robust contrastive learning approach that resorts to pre-trained models (e.g., MoCo-v2) to recall the potential false negatives.

Weaknesses:
1. What are the differences between the existing FN-robust works and the proposed approach? Some works usually first pre-train the models in the first stage and then recall the false negatives in the second stage.
2. Some details are missing. First, the authors does not detail the contrastive learning framework used. Second, how to set the hyper-parameter $K$ for STL and Cifar-10 datasets.
3. I think the comparisons are insufficient and unfair to some extent. First, some newest FN-robust methods (2023) are not compared. Second, the approach adopts additional external models while the existing FN-robust methods usually are trained from scratch.

---

> ### Author Response · Authors · 2024-01-12
> **Review response to Tg1e**
>
> Firstly, thank you very much for your review. We very much appreciate the constructive criticisms provided and hope that we can alleviate the issues which you had with our paper.
>
>
> 1- What are the differences between the existing FN-robust works and the proposed approach? Some works usually first pre-train the models in the first stage and then recall the false negatives in the second stage.
>
> The other approaches use augmented images (distorted view of the instance) with the model under training (not convergence) to find semantic positive pairs, which reduces the accuracy of finding semantic positive pairs, which leads to degenerating the representation learning and slow model convergence. In contrast, we introduced a new search algorithm for finding highly accurate semantic positive pairs using semantic samplers (i.e., pre-trained model and similarity metric) and original images from the dataset because the pre-trained model is more capable of discriminating between classes, especially with original images in the dataset. We described the key differences between our approach and the existing FN works (to the best of our current knowledge in literature) in Sections 1& 2 of our paper.  Our approach's accuracy is 92% in finding the semantic positive pairs, while the other approaches' accuracy does not exceed 57% because of the mentioned reasons. Finding semantic positive pairs with our approach improves the visual representation learning of various self-supervised instance discrimination approaches such (VICReg, SimSiam, DINO, and MoCo-v2).
>
> 2- Some details are missing. First, the authors does not detail the contrastive learning framework used. Second, how to set the hyper-parameter K for STL and Cifar-10 datasets.
>
> Many thanks. We use different frameworks and benchmark datasets to ensure our approach is consistent with instance discrimination approaches. In the main paper experiments, we used momentum contrastive learning as a framework for our approach, and this is mentioned in the section “Experiments and Results” under “Comparing with SOTA Approaches:”. In addition, we described the hyperparameters we used to train our model with a momentum contrastive approach in the training setup. In the “Ablation Study” section, we conducted a further study of our approach with different frameworks. We used SimSiam, DINO, and VICReg frameworks with our method to see how our approach works with non-contrastive learning approaches.
>  Regarding the K value for STL-10 and CIFAR-10 is (10%), and this is mentioned in the subsection “Comparisons on Different Datasets”. We used  K=10% with the ImageNet dataset, so we want to see if we can obtain the same improvement in representation learning with the same proportion on smaller datasets.
>
> 3- I think the comparisons are insufficient and unfair to some extent. First, some newest FN-robust methods (2023) are not compared. Second, the approach adopts additional external models while the existing FN-robust methods usually are trained from scratch.
>
> We have uploaded a revised version of our paper, where we have added 10 new comparisons in Table 3 for the relevant papers published by 2023 that are pre-trained on ImageNet and have followed the standard linear evaluation protocol. We would be delighted to cite additional related papers if the reviewer thinks we have missed any.
>
> In our approach, the task of the external pre-trained model (MoCo-v2) in the semantic sampler **ends** when we find the semantic positive pairs and attach them with the positive pairs to create a new dataset containing both semantic positive pairs (two instances from the same category) and positive pairs (two views for the same instance).  After that, an instance discrimination model such as (DINO, Simsiam, MoCo-v2 ...etc.) is trained from **scratch** on both types of pairs.  We did extensive experiments to prove that our approach improves the representation learning for different instance discrimination frameworks, both contrastive and non-contrastive, by finding highly accurate semantic positive pairs.

---

### Review · Reviewer_LdV9 · 2024-01-03

**Summary Of Contributions:**

This paper studies how to obtain better positive pairs in self-supervised learning (SSL) and proposed a method called SePP-ID. The proposed method use pre-trained SSL models to identify positive pairs in the sampled batch. By employing such method on previous methods such as MoCo-V2, the performance of SSL can be significantly improved.

**Audience:**

Yes

**Broader Impact Concerns:**

No broader impact statement is needed.

**Claims And Evidence:**

Yes

**Requested Changes:**

1) See weakness section.
2) To some extent, the proposed method can be viewed as a special type of distillation method, as the supervision signal is from another pre-trained model. Therefore, it would be better for the author to discuss/compare between the proposed method and some previous works on distillation of SSL models.

**Strengths And Weaknesses:**

Strength:
1) The motivation of mining better positive pairs is interesting and reasonable.
2) The proposed method brings significant improvements on both the non-contrastive and contrastive methods.
3) Configurations such as which pre-trained SSL model to use, k-value, are well-ablated in the experiments.

Weakness:
1) The experiments of adding SePP-ID on stronger baseline such as NNCLR are missing.
2) The experiments of adding SePP-ID on distillation-based SSL such as SWAV and DINO are missing.

---

> ### Author Response · Authors · 2024-01-12
> **Review response to LdV9**
>
> Firstly, thank you very much for your review. We very much appreciate the constructive criticisms provided and hope that we can alleviate the issues which you had with our paper.
>
> 1- The experiments of adding SePP-ID on stronger baseline such as NNCLR are missing.
>
> We view NNCLR as a method in parallel to ours for using semantic positive samples to improve the performance of SSL. Therefore, instead of using the NNCLR framework, we compared our result with the NNCLR in a linear evaluation setting, and the results are shown in Table 3. Also, we used a semi-supervised finetuning setting and the result is presented in Table 4.
>
> 2- The experiments of adding SePP-ID on distillation-based SSL such as SWAV and DINO are missing.
>
> We used the Simsiam framework as an example of knowledge distillation, and the results are reported in Table 8. As suggested, we added SePP-ID with the DINO framework, which is described in Section 4.1.3 in the highlighted text. The results are shown in Table 10.

---

### Review · Reviewer_DfRX · 2024-01-05

**Summary Of Contributions:**

This paper explores an instance discriminative-based self-supervised method. Instead of using augmented images as positive pairs, this paper proposes adopting images with similar semantic content as positive instances. Specifically, the necessary positive pairs are obtained by searching with a pre-trained model, ensuring the correct positive pairs from the start of model training. Experiments are conducted on benchmark datasets, such as ImageNet.

**Audience:**

Yes

**Broader Impact Concerns:**

Na.

**Claims And Evidence:**

Yes

**Requested Changes:**

See weaknesses.

**Strengths And Weaknesses:**

Strengths:

1. The investigated problem is important. Learning effective feature representations from unlabeled data has gained significant attention and has practical applications.
2. The method is well-motivated and implemented. Positive pairs, especially hard positive pairs, play a crucial role in self-supervised learning (SSL), and utilizing a pre-trained model to identify such pairs is a reasonable approach.
3. Experiments are conducted using the ImageNet dataset, which is a widely accepted standard for evaluating computer vision algorithms.

Weaknesses:

1. Limited downstream evaluation. The paper only evaluates the method on a downstream classification task using a linear evaluation protocol. Including more downstream tasks, such as classification under fine-tuning protocol, detection, and segmentation, would strengthen the paper's findings.
2. Are they determined based on ground truth labels? If so, it is important to report the self-supervised results based on the ground truth labels, as well as results obtained through supervised training. This would provide an upper-bound reference and enable a more comprehensive analysis of the proposed method's performance.

---

> ### Author Response · Authors · 2024-01-12
> **Review response to DfRX**
>
> We would like to take this opportunity to thank you for your review of our paper and hope that we can alleviate your concerns below
>
> 1- Limited downstream evaluation. The paper only evaluates the method on a downstream classification task using a linear evaluation protocol. Including more downstream tasks, such as classification under fine-tuning protocol, detection, and segmentation, would strengthen the paper's findings.
>
> In the updated version, we added semi-supervised fine-tuning as another evaluation for our approach. We added two paragraphs (highlighted) on page 9&10 describing the evaluation protocols. The results of using semi-supervised fine-tuning are shown in Table 4.
>
> 2- Are they determined based on ground truth labels? If so, it is important to report the self-supervised results based on the ground truth labels, as well as results obtained through supervised training. This would provide an upper-bound reference and enable a more comprehensive analysis of the proposed method's performance.
>
> We do not use labels to find semantic positive pairs in the dataset. Instead, we aim to propose a scheme to identify semantic positive samples without using labels. We propose a scheme called Semantic Sampler to achieve such a goal (see Figure 4.) We found that using pre-trained self-supervised learning models and a similarity metric is working very well.

---

### Decision · Action_Editor_vJQn · 2024-03-31

**Recommendation:** Accept with minor revision

**Comment:**

The reviewers unanimously recommended leaning accept/accept to the paper upon reading the author rebuttal and additional updates. The AE also appreciates the relatively comprehensive survey in the related-work section. While this work is slightly above the TMLR standard for acceptance, there are weaknesses the authors should try to address:

1) A fair comparison to FNC/NNCLR is needed. Either the authors should show an improvement on top of these two methods as reviewer LdV9 mentioned, or the authors should compare to them on a fair basis. Right now, the reported results from FNC/NNCLR seem to be based on SimCLR, which is a weaker starting point compared to MoCov2. This makes the comparison right now "apples-to-oranges".

2) Downstream evaluation is still lacking. The authors did not do a good job answering reviewer DfRX despite the new SSL results. The authors are highly recommended to consider similar transfer learning experiments as shown in Table 3 of NNCLR. Additional downstream evaluation on segmentation or detection as the reviewer mentioned will also be appreciated.

**Audience:**

This paper studies an important problem in self-supervised representation learning. The paper will be of interests to multiple communities, including visual recognition, representation learning, transfer learning, and metric learning.

**Claims And Evidence:**

This paper proposes an approach to find semantic positive pairs. In particular, a pre-trained model, such as MoCov2, is used to identify semantically identical images to the anchor image using the cosine distance. Since the model is already pre-trained, it avoids making mistakes in selecting positive pairs like FNC and NNCLR. In the paper, the authors show that the proposed method shows higher accuracy than FNC and NNCLR in selecting the right positive pairs.

The proposed method is shown to improve instance discrimination based self-supervised learning tasks by giving stronger positive sample supports. Add the approach on top of MoCov2, the authors show improvements over existing SSL methods. The authors also validated the effectiveness of the proposed approach on other SSL methods, such as SimSiam, DINO and VICReg.

---

> ### Author Response · Authors · 2024-04-17
> **camera version of our paper**
>
> Dear Action Editor,
> We are pleased to submit the camera ready version of our paper. We have added the SimCLR results requested in your point 1) on Table 3  (page 10) and transfer learning on downstream tasks and object detection on Table 5 (page 11) and Table 6 (page 12) respectively; as requested in your point 2). We would like to thank you again for handling our paper so promptly.
> Kind regards,
> The Authors